# Is There a Role for Immunoregulatory and Antiviral Oligonucleotides Acting in the Extracellular Space? A Review and Hypothesis

**DOI:** 10.3390/ijms232314593

**Published:** 2022-11-23

**Authors:** Aleksandra Dondalska, Sandra Axberg Pålsson, Anna-Lena Spetz

**Affiliations:** Department of Molecular Biosciences, The Wenner-Gren Institute, Stockholm University, 10691 Stockholm, Sweden

**Keywords:** oligonucleotide, TLR, sncRNA, endocytosis, broad-spectrum, antiviral agent, nucleolin, virus entry, immunoregulation, RNA therapeutics

## Abstract

Here, we link approved and emerging nucleic acid-based therapies with the expanding universe of small non-coding RNAs (sncRNAs) and the innate immune responses that sense oligonucleotides taken up into endosomes. The Toll-like receptors (TLRs) 3, 7, 8, and 9 are located in endosomes and can detect nucleic acids taken up through endocytic routes. These receptors are key triggers in the defense against viruses and/or bacterial infections, yet they also constitute an Achilles heel towards the discrimination between self- and pathogenic nucleic acids. The compartmentalization of nucleic acids and the activity of nucleases are key components in avoiding autoimmune reactions against nucleic acids, but we still lack knowledge on the plethora of nucleic acids that might be released into the extracellular space upon infections, inflammation, and other stress responses involving increased cell death. We review recent findings that a set of single-stranded oligonucleotides (length of 25–40 nucleotides (nt)) can temporarily block ligands destined for endosomes expressing TLRs in human monocyte-derived dendritic cells. We discuss knowledge gaps and highlight the existence of a pool of RNA with an approximate length of 30–40 nt that may still have unappreciated regulatory functions in physiology and in the defense against viruses as gatekeepers of endosomal uptake through certain routes.

## 1. Introduction

Recent advances have identified a pool of single-stranded oligonucleotides (ssONs) (either DNA or RNA) with features resulting in the capability to inhibit a broad range of enveloped viruses by binding to, or shielding, viral entry receptors [1,2,3,4,5]. These oligonucleotides do not target a specific sequence (hence, they are not antisense or sequence mimics) and can be administered in vivo without using cell-penetrating peptides or other delivery systems. One such class of ssDNA, oligonucleotides are the nucleic acid polymers (NAPs), which act by facilitating interactions with un-complexed amphipathic alpha helices (Reviewed in [1]). These NAPs are typically 40-mer phosphorothioate oligonucleotides and have been shown to interact with HIV-1 gp41 [6], the surface glycoprotein of lymphocytic choriomeningitis virus [7], prion proteins [8], and hepatitis delta antigen [9] in a sequence-independent manner. However, some NAPs act in other steps of the viral life cycle such as REP 2139, which was reported to inhibit secretion of Hepatitis B surface antigen (HBsAg) [10,11]. The proposed mechanism of action of REP 2139 is that blocked replenishment of HBsAg in the circulation facilitates host-mediated viral clearance [10]. A historical background with more in-depth introduction to the chemistry of nucleic acids therapeutics was recently published [12].

Another set of ssDNA, which are also acting in a sequence-independent manner, was shown to inhibit TLR3 activation [13,14,15]. It was revealed that this class of ssONs inhibits TLR3 activation by temporarily inhibiting clathrin-mediated endocytosis, thereby preventing the uptake of the TLR3 ligand dsRNA [15]. Notably, the inhibitory concentration (IC)_50_ of ssONs with a capacity to inhibit TLR3 activation, is around 125 nM [15]. In addition, this class of ssONs were shown to inhibit infection of viruses such as respiratory syncytial virus (RSV) [3] and HIV-1 [5] in low nM concentrations. These potent activities of ssONs (in the low nM range) make us wonder whether there could be a role for naturally occurring oligonucleotides in the extracellular space and if so, which are the potential sources of such oligonucleotides? This review will focus on TLR activation requiring endocytic uptake of ligands and mechanisms on how such TLR activation can be regulated. Moreover, we review data showing that certain oligonucleotides possess antiviral activity without requiring delivery systems for intracellular uptake. Lastly, this review will provide information on the development of therapeutic approaches utilizing ssONs that act in the extracellular space.

## 2. Characteristics of Therapeutic Oligonucleotides

### 2.1. ASO

Single-stranded antisense oligonucleotides (ASO) are small (approx. 15–30 nt long), synthetic, nucleic acid polymers that often contain various chemical modifications to improve their therapeutic efficacy (such as stability and cellular uptake). ASO sequences are designed to bind to their target RNA by utilizing Watson and Crick base pairing in order to modulate splicing, affect translation initiation, or promote recruitment endonucleases such as RNase H to increase degradation of the disease-causing RNA. The first approved ASO was fomivirsen, which was used to locally treat cytomegalovirus (CMV) infection [12]. There are currently two widely used classes of ASO; the “gapmers” (utilizing RNase H) and the “splice-switching” (steric block) ASO. The current ASO that utilize the enzymatic activity of endogenous RNase H follow a “gapmer” pattern, meaning that a middle region of 6–10 DNA bases is surrounded by RNA bases that promote target binding. The RNase H recognizes the RNA–DNA heteroduplex substrates that are formed when the DNA-based oligonucleotide binds to the cognate mRNA transcript and causes RNA degradation. RNase H can cleave the RNA strand of a DNA–RNA duplex in both the nucleus and the cytoplasm. Hence, the function of gapmers is reliant on intracellular uptake of the ASO. ASOs typically enter cells via endocytosis and after uptake into early endosomes, they have to cross the endosomal lipid barrier to access their target in the cytoplasm and/or nucleus [16]. However, the mechanism for so called “endosomal escape” is poorly understood. Nevertheless, there are currently three different gapmers approved by the FDA and/or EMA mipomersen, inotersen, and volanesorsen, and they are all 20-mers (Table 1).

High affinity, sterically blocking ASO, are designed to bind to their target mRNA thereby masking specific sequences within their target transcript, which causes interference with RNA–RNA or RNA–protein interactions. ASO approaches often modulate alternative splicing, which leads to selective exclusion or retainment of a specific exon(s). Splice correction methods have often been used to restore the translational reading frame in order to salvage the target protein production [17]. So far, five different splice-switching ASO have been approved by the FDA; eteplirsen (30-mer), nusinersen (18-mer), golodirsen (25-mer), viltolarsen (21-mer), casimersen (22-mer) (Table 1).

### 2.2. siRNA

The antisense (or guide) strand of a double-stranded small interfering RNA (siRNA) also binds its target mRNA by Watson–Crick base pairing and is likewise reliant on uptake into cells. The other strand is designated by the passenger or sense strand. siRNA work by guiding the Argonaute (AGO) 2 protein, as part of the RNA-induced silencing complex (RISC), to the complementary target transcripts. Exact base pairing between the siRNA and the target transcript results in cleavage of the antisense strand, leading to gene silencing [17]. AGO2 protein and the RISC complex have specific structural requirements that the oligonucleotide must possess in order to bind, which limits the extent of chemical modifications that can be introduced and reflects in the stability and cellular uptake [16]. As of October 2022, five siRNAs have received FDA approval; patisiran, givosiran, lumasiran inclisiran, and vutrisiran (Table 1).

### 2.3. Others

Aptamers are ssONs typically approx. 20–200 nt that are folded into defined secondary structures and act as ligands that bind to target proteins in a similar way as antibodies. They can be generated by using an in vitro evolution methodology called SELEX (systematic evolution of ligands by exponential enrichment) [17]. Currently, the only approved aptamer is pegaptanib, which is an RNA-based aptamer that targets the VEGF-165 vascular endothelial growth factor isoform. Pegaptanib is used for its anti-angiogenic properties for the treatment of neovascular age-related macular degeneration [19,20]. There are currently more than 40 clinical trials listed at https://clinicaltrials.gov (accessed on 20 October 2022) concerning investigations of aptamers.

The approved drug defibrotide is produced from porcine mucosa and is composed of a mixture of ssDNA and dsDNA (hence phosphodiester (PO)-DNA) of different sizes [22]. The mechanism of action is poorly defined as of yet, however, it is not strictly sequence dependent [23]. The use of naturally occurring PO-DNA in a drug like defibrotide, indicates that there is still a considerable amount to be revealed, in terms of how pools of oligonucleotides contribute to the regulation of normal physiological mechanisms.

The approved medicines shown in Table 1 demonstrate the significance of oligonucleotides in current medical practice. The future importance of oligonucleotide-based therapies is further demonstrated by the notion that there are more than 250 clinical trials listed in https://clinicaltrials.gov (accessed on 20 October 2022) utilizing oligonucleotide therapeutics and numerous studies have been performed over the years. The concepts in progress include different types of CpG-oligodeoxynucleotide (ODN), NAPs, micro RNA (miRNA) (using mimics or anti), CRISPR-based technologies, small nucleolar RNA (snoRNA), Ribozymes, long non-coding RNA (lncRNA) (using antagoNATs or small activating RNA), and of course mRNA (vaccines, VEGF cardiac regeneration) approaches that were boosted during COVID-19 with the success of these vaccines [12,16]. Hence, these are exciting times for therapeutic oligonucleotides and there will likely be many more approved within the next years. Yet, there are still many unanswered questions in terms of optimizing delivery systems into cells of different organs and also how to exploit oligonucleotides that act on the cell surface, circumventing the obstacles of intracellular delivery. The field of therapeutic oligonucleotides is naturally directly linked to the identification and functional characterization of new non-coding RNAs and also how the immune system is reacting to oligonucleotides depending on their subcellular and/or extracellular location.

## 3. The Expanding Identification of Small Non-Coding RNAs

In the last decades, the existence of functional regulatory sncRNA has been revealed in all kingdoms of life, e.g., from bacteria and archaea to various eukaryotes due to transcriptome-wide studies and advances in high-throughput sequencing technologies [36]. Nevertheless, technical challenges still remain to accurately discover and measure the plethora of such sncRNA. There is a substantial amount of information about siRNAs, miRNAs, and PIWI-interacting RNAs (piRNAs) which act by base pairing to their respective RNA and/or DNA targets to exert RNA-silencing effects (such as post-transcriptional mRNA cleavage, decay or translational repression and transcriptional silencing) via AGO or PIWI. PIWI is an abbreviation of P-element Induced WImpy testis in Drosophila and are highly conserved RNA-binding proteins, which are present in both plants and animals. There are also emerging data on other non-canonical sncRNA that are often approx. 13–200 nt and derive from longer RNAs such as transfer RNA (tRNA), ribosomal RNA (rRNA), Y RNA (yRNA), small nuclear RNA (snRNA), snoRNA, vault RNA (vtRNA) and even mRNA as well as lncRNA. Several authors ascribe sncRNAs in the range of approx. 15–50 nt [37,38,39]. Similar to many non-coding RNAs in history, these emerging sncRNAs were initially considered to be random degradation products without a defined functional role, but increasing evidence shows their functional regulatory role in both health and diseases. There are links to cancer, immunity, viral infection, neurological diseases, stem cells, retrotransposon control, and epigenetic inheritance (recently reviewed in [38]).

The definition of “small” non-coding RNA is relatively subjective in different contexts and is likely to be more streamlined in the future when more insights are gained. Nevertheless, the different sizes in terms of numbers of nt are important to keep track of as certain features are linked to the length, and the current sequencing techniques used to identify sncRNA are often biased based on sizes of captured RNA [38]. For example, many protocols were focusing on sequencing miRNA and siRNA by using a pre-size selection with a cut-off <30 nt RNA by recovering RNA from electrophoresed gels and thereby prevented the discovery of sncRNAs, which were more than 30 nt [38]. The RNA size selection was later extended to approx. 45 nt, which can include PIWI-interacting RNAs (piRNA) and also led to the discovery of tRNA-derived-small RNA (tsRNA) and yRNA-derived-small RNA (ysRNA) found at 30–40 nt (Table 2 nomenclature of fragments in part from [38]). In addition, it was recently realized that many non-canonical sncRNAs carry various RNA modifications, some of which can prevent their detection by traditional RNAseq [38]. Hence, there is a need to develop novel sequencing techniques that can directly sequence RNA and simultaneously identify modifications, which can be as many as 150 types [40].

Another major challenge concerns the subcellular spatial compartmentalization including measurements in the extracellular space of sncRNAs. Great advances have been made to spatially map the transcriptome based on in situ hybridization, either through multiplexed imaging [41], or sequencing [42], and the single-cell resolution is likely to be further improved. However, these methods are optimized for long mRNA, whereas the short length of sncRNAs limits nucleic acid probe design resulting in binding to multiple targets [38]. It is going to be exciting times to follow the development of paradigm-changing tools allowing for the spatial and compartmentalized high-resolution discovery of sncRNAs in various tissues in health and disease. It was reported that the profile of ncRNAs can be changed upon virus infections [43,44,45]. For instance, RSV infection can induce the upregulation of certain tRNA fragments that enhance the replication of the virus by affecting the antiviral response [43]. It is conceivable that sncRNAs possess key functions both inside and outside of cells if released from dying cells [46], extracellular vesicles (reviewed in [16]), or hypothetically directly exported from cells. Notably, there are highly conserved cytoplasmic non-coding yRNAs, which range in size from 70 to 115 nt, and typically form Ro ribonucleoproteins (RNPs) by binding to Ro60 and La proteins. These RNPs are often targeted by the immune system in autoimmune diseases, pointing towards an Achilles heel to discriminate self-nucleic acids from foreign ones [46,47]. Although Table 2 is very brief and will require updates on definitions of different types of sncRNAs, which is best performed by the sncRNA field of experts, it points out some key differences in terms of length intervals. siRNA, miRNA, piRNA, and also other types of relatively short ssONs are working by base pairing and gene silencing using AGO proteins or PIWI. The somewhat longer yet highly abundant pool of ssONs of approx. 30–40 nt do not have the features to be able to work via AGO or PIWI and may not necessarily act in a sequence-dependent manner. Notably, fragments derived from lncRNA and mRNA may also fall into the category of 30–40 nt [39].

## 4. Endosomal Immune Receptors Recognizing Nucleic Acids

The innate immune system employs an array of pattern recognition receptors (PRRs) to recognize potentially dangerous microbes and particles. Accordingly, these groups of receptors are specific for highly conserved features of foreign invaders termed pathogen-associated molecular patterns (PAMPs) or host-derived molecules called danger-associated molecular patterns (DAMPs). Infection and tissue damage can induce a rapid inflammatory response due to the activation of the innate immune system via PRRs which recognize different PAMPs and DAMPs. PRRs are found in particular locations in the cell corresponding to their ligand specificity; hence, the receptors that detect extracellular PAMPs are expressed on the surface of the cell, while PRRs found in the cytosol, sense microbial infections and include NOD-like receptors, RIG-like receptors (RLRs, (RIG-1, MDA-5, and DExD/H-box helicases)), and cytosolic DNA sensors [55,56,57]. Other PRRs, including certain TLRs, are expressed in endosomal compartments and can recognize foreign nucleic acids or microbes that have been endocytosed by the cell [56,57]. Recognition and binding to cognate PAMP or DAMP ligands by a specific PRR lead to the activation of a signaling cascade that culminates in a coordinated intracellular innate immune response designed to control infection or heal a stress response; this includes type I and III interferons and pro-inflammatory cytokines and chemokines, as well as factors that modulate the expression of innate genes that promote an antiviral cellular state. These genes encode factors with direct antiviral action or genes that modulate the metabolic or cell cycle state, which leads to restricted viral production. Secreted interferons and cytokines will augment the innate response locally and recruit immune cells to the site, which altogether results in the expression of a plethora of interferon-stimulated genes (ISGs), facilitating additional antiviral activities [57].

TLRs belong to a conserved family of transmembrane glycoprotein receptors, for which humans possess 10 genes. All TLRs are similarly organized with an extracellular leucine-rich-repeat domain, a transmembrane domain, and a cytosolic Toll-IL1R (TIR) domain that extends into the cytosol and mediates downstream signaling after receptor activation [58]. TLRs are highly expressed in many immune cells and endothelial cells as well as epithelial cells, and keratinocytes. However, each cell type expresses a distinct repertoire of TLRs [59]. As shown in more detail in Figure 1, the activation of TLRs triggers a critical immune response for host defense that is specific for the respective TLR and induces signaling that leads to the generation of different effector molecules including interferons and pro-inflammatory cytokines. However, dysregulation of signaling or ligand recognition by TLRs is associated with the pathogenesis of inflammatory and autoimmune diseases [60]. Hence, there is a distinct requirement to downregulate the innate response to avoid chronic inflammatory responses [56,61]. Yet, the response has a crucial role in not only responding to the pathogens per se, but also taking part in the healing response and clearance of DAMPs.

Foreign nucleic acids are specifically recognized by RLRs, cytosolic DNA sensors, and a subgroup of TLRs [57]. The subgroup of TLRs that can recognize nucleic acids consists of TLR3, 7, 8, and 9, which all primarily reside in endosomal compartments unlike other nucleic acid sensors, which are found in the cytosol. In general, these TLRs can signal from endosomes by binding to structures that are only accessible once they are taken up and degraded or part of a virus utilizing a certain endocytic pathway to enter the cell. Notably, each endosomal TLR recognizes a specific type of nucleic acid [57].

TLR3 recognizes dsRNA after uptake through clathrin-mediated endocytosis [62]. The sources of dsRNA recognized by TLR3 have been reported to be long dsRNA [63,64] but also ssRNA with stem-loop structures [65]. Although TLR3 seems to have limited sequence specificity, the affinity increases proportionately to dsRNA length [63,64]. The dsRNA has been suggested to bind through electrostatic interactions and hydrogen bonds and to require a minimum length of 40–50 bp for TLR3 activation [63,64,66,67].

TLR7 and TLR8 detect ssRNA but also require binding of RNA degradation products for signaling [68,69,70,71,72]. Hence, TLR7 and TLR8 have two ligand binding sites that have to be occupied for activation and the presence of both products will activate TLR7 or TLR8 signaling. TLR7 has been shown to have a preference for binding to guanosine at the first binding site and recognizes a 3-mer motif embedded within long stretches of polyU-ssRNA at the second site, in which the critical residue is uridine [68,69]. TLR8, on the other hand, has a preference for binding to uridine at the first site and has been suggested to recognize GU-rich ssRNA at the second site [70,72].

TLR9 responds to unmethylated CpG motifs, which consist of a central unmethylated CG dimer flanked by 5′ purines and 3′ pyrimidines, present at high frequency in bacteria [55]. TLR9 is highly expressed in plasmacytoid dendritic cells but can also be detected in B cells and keratinocytes [73,74]. Notably, there are differences between species in TLR9 expression as rodents express TLR9 also in macrophages and myeloid-derived dendritic cells. Therefore, studies in mice may overestimate the activity of TLR9. Four distinct classes of CpG have been identified and used in multiple clinical trials as adjuvants in vaccines or cancer treatment, recently reviewed in [73]. The K-type ODNs (also referred to as B-type) contain 1–5 CpG motifs and have a PS backbone. The D-type ODNs (also referred to as A-type) typically express a single CpG motif flanked by palindromic sequences enabling the formation of a stem-loop structure. The central nucleotides in the D-types are phosphodiester (PO) while the ends are capped with PS (polyG motifs at 5′, 3′or both ends).

The activation of nucleic acid sensing TLRs triggers the induction of numerous molecules involved in the innate immune response. These include pro-inflammatory cytokines such as IL-6 and TNF-α that are highly produced by macrophages, conventional dendritic cells, and B cells, and type I interferons (IFNs), which are secreted in high amounts by plasmacytoid dendritic cells [75]. As shown in more detail in Figure 1, the binding of agonists to all TLRs, except TLR3, results in a signaling cascade depending at least in part on adaptor myeloid differentiation primary response 88 (Myd88) [58,75]. While TLR7 and TLR9 downstream signaling is completely reliant on MyD88, TLR3 solely utilizes the TIR domain-containing adaptor protein-inducing interferon-beta (TRIF). The compartmentalization of nucleic-acid sensing TLRs to endosomal compartments is crucial to their specialization as it allows the innate immune system to detect non-self, internalized nucleic acids, and additionally assures protection from their activation by self-nucleic acids and subsequent induction of autoimmunity [61].

For therapeutic oligonucleotide design, it is preferred to evade immune recognition since unmodified oligonucleotides can activate RIG-I and PKR, which detect dsRNA in the cytoplasm. However, it has been shown that certain chemical modifications such as 2′-OMe-modifications of uridine and guanidine residues in siRNA aid in immune evasion [16]. Nevertheless, there is still a knowledge gap in our understanding of immunogenic properties when designing therapeutic nucleic acids. It can be noted that oligonucleotides with neutral backbones have so far not been implicated in immune activation [16].

TLR3 is expressed in the endosomal compartments of diverse leukocytes such as myeloid dendritic cells, natural killer cells, and macrophages as well as non-immune cells including fibroblasts, endothelial cells, epithelial cells, keratinocytes, and neurons [14,59,74,76,77,78,79]. Due to the notion that TLR3 recognizes dsRNA, which can be found in some viral genomes and is produced during viral replication cycles, it constitutes a key component in the recognition and clearance of certain viral infections [80,81]. The activation of TLR3 ensues after suitable ligand binding and leads to the interaction and dimerization of TLR3 extracellular domains [60]. This is followed by signal transduction which leads to the activation of nuclear factor kappa-light-chain-enhancer of activated B cells (NF-κB) and the production of pro-inflammatory cytokines. Moreover, TRIF-mediated signaling can lead to the production of type 1 IFNs through the phosphorylation and activation of interferon regulatory factor 3 (IRF3). [60,82]. The release of pro-inflammatory cytokines and type I IFN by activated or infected cells is crucial in initiating inflammatory and adaptive antiviral responses [60].

TLR3 activation is not limited to agonists of viral origin and can be induced by dsRNA released by damaged tissue or the widely used synthetic dsRNA analog, polyI:C. While TLR3 activation is important for an efficacious immune response towards viral infection, in the course of which dsRNA accumulates throughout viral replication, it has the potential to initiate undesirable effects. In principle, endogenous nucleic acids can trigger TLR3-dependent immune responses, thereby contributing to inflammatory pathologies and autoimmunity [83,84,85]. Further, excessive TLR3 activation has been shown to increase pathology in some viral infections [86,87,88] and it has been implicated in undesirable outcomes such as virus-induced asthma [89]. However, damage of the lung does not necessarily have to be due to infection, but can also be enhanced by sterile cell death, e.g., during oxygen treatment of patients with acute respiratory distress syndrome [90], organ transplant complications, cardiovascular disease and diabetes or autoimmune reactions such as arthritis [76,77,91,92,93]. All nucleic acid sensing endosomal TLRs have been implicated in asthma, wherein TLR3 has been associated with induction, and TLR7, TLR8 and TLR9 are associated in disease exacerbation [94,95]. Human TLR3-mediated immunity, at least in some individuals, is pivotal for protection against HSV-1 infection in the CNS [81] and there is an increasing number of reports showing augmented disease severity in patients with TLR3 deficiency [96,97,98], otherwise, TLR3 signaling is remarkably redundant.

Altogether, there is extensive support in the literature that over-reactivities involving the recognition of nucleic acids, which are taken up into endosomes and trigger TLRs, have a key role in the pathogenesis of several autoimmune and allergic reactions [14,46,47,80,99].

## 5. Discovery of SOMIE

We previously showed that a 35 mer CpG ssON (B-type) could inhibit TLR3 signaling in primary human monocyte derived cells (moDC) that express TLR3/4/8, but lack TLR7/9 [14]. This was a serendipity finding discovered when we were combining different TLR agonists and measured moDC activation in order to disclose combinations with additive and/or synergistic effects. As the moDC lacked TLR9 expression, we realized that the inhibitory effect was not dependent on TLR9. Consequently, removal of the CpG motifs and replacement with another nucleotide did not alter the inhibitory effect [14]. We provided evidence that polyI:C–induced DC maturation was inhibited by ssON on all levels investigated: upregulation of maturation markers, production of type I interferons and proinflammatory cytokines, and phosphorylation of vital transcription factors (NFkB, IRF-3). We also showed that polyI:C–induced cytokine production in the airways of cynomolgus macaques was significantly blocked by ssON [14].

As a next step, we further elucidated mechanisms involved in the inhibition of TLR3. It was previously found that ssONs containing TTAGGG motifs to mimic telomeric DNA had a general immunosuppressive effect because of inhibition of STAT signaling [100], while ssONs with polyG motifs inhibited TLR9 activation (and in some cases also TLR7 and TLR8), with the suggested mechanism of competitive antagonism [101,102,103]. However, the ssONs discovered in [14] did not have such motifs.

We synthesized a panel (varying sequences, modifications and length) of ssON to identify the requirements for the inhibition of dsRNA-mediated activation [15]. We discovered that ssON not only inhibited TLR3 activation, but also inhibited the activation of TLR7 in PBMC making it unlikely that it was a direct TLR3 antagonistic mechanism. Further, we found that ssON modulated TLR4 activation that was dependent on endosomal uptake, while leaving TLR4 signaling from the plasma membrane unaffected in human moDC [15]. Extracellular cargo destined for TLR3/4/7 signaling endosomes is taken up by endocytosis. We therefore investigated whether the inhibition of TLR activation was due to decreased uptake of ligands into endosomes. Indeed, we provided evidence that certain ssON temporarily downregulate clathrin-mediated endocytic activity, thereby revealing a gate keeping mechanism for TLR3/4/7 activation. We termed the ssON-mediated interference of endocytosis SOMIE and showed that both single-stranded RNA and DNA, but not dsDNA, have this capability [15] (Figure 2).

The endocytic inhibition was concentration dependent and not strictly sequence dependent. However, there was a length requirement of at least 25 nt. It is conceivable that not all cell types respond to the SOMIE effect, but we have shown inhibition of TLR3 activation in moDC, keratinocytes, epithelial cells and fibroblasts [14]. Whole cell proteomic and transcriptomic analyses revealed that there were remarkably few cellular changes occurring in moDC after SOMIE [15]. We further provided evidence that ssON modulate TLR3 activation in vivo in macaques [15]. This opens possible novel therapeutic avenues for autoimmunity involving endocytosis-dependent TLR3/4/7 activation. Our findings that either ssDNA or ssRNA of at least 25 nt have the capacity to temporarily shut down clathrin-mediated endocytosis opens up intriguing questions in host–viral interactions, RNA biology, and autoimmunity. Our data have implications for questions in several fields such as (1) viral endocytosis, (2) cellular uptake of oligonucleotide therapeutics and endogenous sncRNA (13–24 nt), (3) role of stabilizing modifications of endogenous 25–40 nt pool of RNA, (4) immune regulatory function of non-coding RNAs, and (5) development of ssON-based therapeutics acting in the extracellular space.

## 6. ssONs Acting as Attachment/Entry Inhibitors of Viruses

Clathrin-mediated endocytosis is a key step for cellular entry of many viruses that can lead to the triggering of TLRs located in endosomes [104]. The development of entry inhibitors aligns with a WHO incentive to target virus-associated host factors, which are theoretically less prone to the development of resistance. As these host factors are employed by multiple viruses, there is also the possibility to achieve a broad antiviral coverage. Targeting virus entry is appealing [105] as it blocks the first step in the viral life cycle and shuts off subsequent replication and pathogenic processes. We therefore reasoned that it would be prudent to investigate whether ssONs with the capacity to inhibit endocytosis could function as an antiviral agent. We indeed obtained data showing reduced influenza A (H1N1) infection in vitro in human cells and in a murine in vivo challenge model after ssON treatment [2]. We next investigated the capacity of ssONs to inhibit RSV infection [3,4]. We found that the ability to block RSV infection was dependent on the length of the oligonucleotide, but not strictly sequence dependent as a selection of ssONs between 25–35 nt have the ability to inhibit RSV infection. Furthermore, we discovered that inhibitory ssONs of either DNA or RNA origin effectively inhibited RSV. We found that the effect to inhibit RSV was not dependent on the PS-modification and PO-ssON could also inhibit RSV infection, although not as efficiently as the stabilized version. Notably, we synthesized a selection of six different sncRNAs derived from yRNA, rRNA, or tRNA, which were identified in the bronchoalveolar lavage of healthy individuals. These sequences were randomly selected from a published data set [106], but they were all in the range of 30–40 nt long. We found that these stabilized sncRNAs also possess the capacity to inhibit RSV infection in a similar low nM range as the “parent” 35-mer ssON [4]. These findings further highlighted the importance of oligonucleotide length, but not necessarily an exact sequence or chemical modification, and revealed that this inhibitory effect could be a naturally occurring phenomenon. We also demonstrated that the likely mode-of action governing the inhibitory effect was acting at the viral entry step by preventing the viral binding to nucleolin [3,4].

## 7. Nucleolin Is a Binding Partner for ssONs

Nucleolin is a nucleic acid-binding protein that exists abundantly in the cell nucleus, but it is also present in the cytoplasm and the cell membrane [107]. Nucleolin is a multifunctional protein and plays key functions in processes such as chromatin remodeling, transcription of rRNA, rRNA maturation, ribosome assembly, and ribosome biogenesis [107]. Nucleolin can bind to either DNA or RNA and was also reported to encompass DNA and RNA helicase activity [107]. It has been indicated as a shuttling protein present in endosomes and to be responsible for transporting proteins between the nucleus, cytoplasm, and the cell surface [108,109,110]. Notably, a recent study showed that nucleolin located on the surface of murine DC bound directly to A-type CpG ODN, B-type CpG ODN, and polyI:C and promoted their internalization. In human DCs, nucleolin also contributed to the binding and internalization of both type A and B CpG ODNs [111]. Intriguingly, nucleolin has been reported to be involved in viral attachment and/or entry of not only RSV [112,113], but also other viruses such as HIV-1, HSV-2, influenza, parainfluenza type 3, enterovirus 71, Crimean–Congo hemorrhagic fever virus, adeno-associated virus type 2, coxsackie B virus, Seneca Valley virus [107,114,115,116].

Nucleolin is currently the only candidate that has fulfilled all requirements to be defined as a functional receptor for RSV. It was demonstrated that pre-treating cells with a nucleolin antibody reduced the infection and pre-treating the virus with soluble nucleolin before infection similarly reduced infection. Furthermore, silencing nucleolin expression by using siRNA, significantly reduced the RSV infection. Additional support for nucleolin as a functional RSV receptor was obtained after the induction of nucleolin expression in normally non-permissive cells, which enabled RSV infection [112,113]. Further studies have shown that by utilizing the DNA aptamer AS1411, which binds nucleolin located on the cell surface, RSV infection can be inhibited in epithelial cell lines and in vivo in mice and cotton rats [117]. The RSV F protein has been shown to be the interacting partner of nucleolin [112] and a recent study showed that the F protein binds specifically to the RBD1,2 (RNA binding domain 1,2) binding site of nucleolin [113].

Several studies indicate that nucleolin is part of a multicomplex consisting of several proteins, which participate in the binding and entry of RSV [113,118]. Furthermore, studies have suggested that RSV facilitates its own uptake by triggering nucleolin to translocate to the surface of cells upon viral binding [119]. A recent report showed that the interaction of the RSV F protein with IGF1R triggered the activation of protein kinase C zeta, which resulted in the recruitment of nucleolin to the cell surface, whereby nucleolin aided in viral entry [120] (Figure 3, left). There is still a lack in our knowledge on which viral-cell binding partners are involved in triggering translocation of nucleolin to the cell surface upon contact with other viruses [116]. It also remains to be further established as to what extent nucleolin is required for the viral entry of other viruses. It is conceivable that there might be differential susceptibility to entry inhibitors targeting nucleolin depending on the virus. Our finding that ssONs with the length of 30–40 nt have the capacity to interfere with nucleolin is intriguing and opens up for additional testing of other viral families reliant on nucleolin during the attachment and/or entry step (Figure 3, right). Notably, we were able to reduce RSV infection in vivo in a murine model [3]. We also studied immune responses occurring locally in the lungs in RSV-challenged mice with or without ssON treatment and found that the ssON treated mice displayed a more profound upregulation of ISGs using the Nanostring technology.

We used qPCR to validate that ssON treatment indeed upregulated expression of ISGs such as *Stat 1*, *Stat 2*, *Ccl2*, and *Cxcl10* [3]. Hence, it is conceivable that blocking one viral entry pathway may force viruses to enter via another uptake route, which hypothetically can trigger cytoplasmatic nucleic acid sensors, which are effective inducers of ISGs. Future studies are required to investigate the cellular and molecular responses occurring locally in the lungs upon ssON treatment, especially as it possesses both antiviral and immunomodulatory properties. This is especially pertinent in an in vivo context as several cell types are involved, both resident and recently infiltrating cells, in a finely tuned and orchestrated immune response.

The finding that we may naturally have sncRNAs in, for example, bronchoalveolar lavage with antiviral potential [4] hints towards a naturally occurring mechanism, although it is conceivable that the PS-modifications may strengthen and prolong the effects.

## 8. Therapeutic Approaches of ssONs Acting in the Extracellular Space

TLR3 is a key receptor for the recognition of dsRNA and the initiation of immune responses against viral infections. However, hyperactive responses can have adverse effects, such as virus-induced asthma but also in other diseases. An over-reactive TLR3 signaling driven by viral, or endogenous dsRNA from dying cells, has been implicated as a common driver in several diseases including viral infections [86,87,88], acute respiratory distress syndrome [90], asthma, organ transplant complications, cardiovascular disease, and diabetes or autoimmune reactions such as arthritis [76,77,89,91,92,93]. Experimental evidence supporting TLR3 involvement was often demonstrated by a diminished response in TLR3 knock-out mice and/or exacerbation of the inflammatory condition by polyI:C. As TLR3 is primarily located in the endosomes and is activated (in an acidic environment) upon the binding of dsRNA taken up from dying cells or virus infected cells, it is not an easy target for small molecule inhibitors or antibody-based therapies. New strategies for drug development may therefore be required to prevent TLR3-mediated pathology.

Our results demonstrate that TLR3-triggered immune activation can be modulated by the 35-mer ssON and provide evidence of dampening proinflammatory cytokine release in the airways of cynomolgus macaques and locally in the skin. Injection in the skin led to a reduction in IL-6 (pro-inflammatory) production and induction of IL-10 (“anti-inflammatory”) [15]. In addition, the 35-mer ssON ameliorates certain itch in vivo in mice and reduces mast cell degranulation [121]. These findings may open novel perspectives for clinical strategies to prevent or treat inflammatory conditions exacerbated by TLR3 signaling.

We used well-known PS-modifications of the ssONs in the antiviral and immunoregulatory experiments, which enhance the stability of nucleic acid drugs by nucleus-mediated degradation (recent review on different modifications [122]). Additionally, the PS backbone confers protein-binding properties enhancing binding to plasma proteins and cell-surface proteins involved in cellular attachment and/or facilitating uptake into cells [123]. Numerous clinical trials have been conducted with PS-stabilized oligonucleotides (PS-ONs) over the past two decades from which conserved class behaviors have been established. After administration by intravenous infusion or subcutaneous injection, PS-ONs are rapidly cleared from the blood (half-life <1 h), concomitant with accumulation in peripheral organs, mostly liver and kidney. They are relatively resistant (compared with PO-ONs) to nuclease-mediated degradation, but degrade slowly over time with the primary route of elimination via the kidney [124]. Thus, further drug development of PS-stabilized ssON as a broad-spectrum antiviral agent active against RSV [3,4], influenza A [2], and HIV-1 [5], and hypothetically blocking nucleolin-dependent entry driven by various glycoproteins of other pathogenic viruses for which there is still a significant medical need, is highly warranted. It is conceivable that the antiviral mechanism acting in vivo is more complex than inhibition of viral binding to cellular surfaces as differential immune responses may also be evoked that may take part in the defense [3]. Hence, careful pharmacokinetic and formulation studies are required to design future animal studies in combination with mechanistic studies to reveal how antiviral 35-mer ssON´s act in vivo.

## 9. Conclusions and Future Prospects

Based on recent advances and progress in the field, there is little doubt that additional nucleic acid-based therapeutics will be approved within the coming years. Currently, the majority of oligonucleotide-based therapeutics are centered on base pairing to target a specific sequence to achieve, for example, gene silencing (Table 1). Nevertheless, the expanding field of non-coding RNA biology also teaches us about the plethora of different RNA fragments of which some may exert functions that are non-sequence specific. It is conceivable that they may have a secondary structure required for RNA–protein binding and modifications to increase stability and/or increase affinity. There is a need to develop techniques that can include sequencing of RNA modifications, and preferentially high-resolution spatial techniques, which can reveal the subcellular and/or extracellular location of the non-coding fragments.

There is currently a lack in our understanding about the functions of the pool of non-coding RNA with the approximate length of 30–40 nt. Recent years have explored the functions of shorter fragments such as miRNA, siRNA, and piRNA but there is a still much to reveal concerning the abundant pool of 30–40 nt that can be found to be derived from different RNA such as tRNA, Y RNA, rRNA, mRNA, and lncRNA. One hypothetical view of the 30–40 nt pool is that they confer a “buffering” system to regulate the uptake of endocytic cargo into cells lining (keratinocytes, epithelial, and endocytic cells) and patrolling (classical DC) the body for the detection of pathogenic intruders.

We have provided evidence that some ssONs have the capacity to temporarily inhibit certain endocytic pathways [15] and have provided evidence of their immunoregulatory functions [121] in experimental conditions based on endocytic uptake and triggering of endosomal TLR3 [14,15]. In addition, this pool of 30–40 nt RNA may also have a role in preventing entry of certain viruses [2,3,4]. There is still a gap in our understanding to what extent they exert broad-spectrum antiviral capacities and if additional immunological properties are associated with their antiviral effects. More knowledge on how the immune system is dealing with extracellular oligonucleotides of different lengths will aid in the design of ASO therapeutics and may also open up the field for a new type of immunoregulatory and/or antiviral therapies.

## Figures and Tables

**Figure 1 ijms-23-14593-f001:**
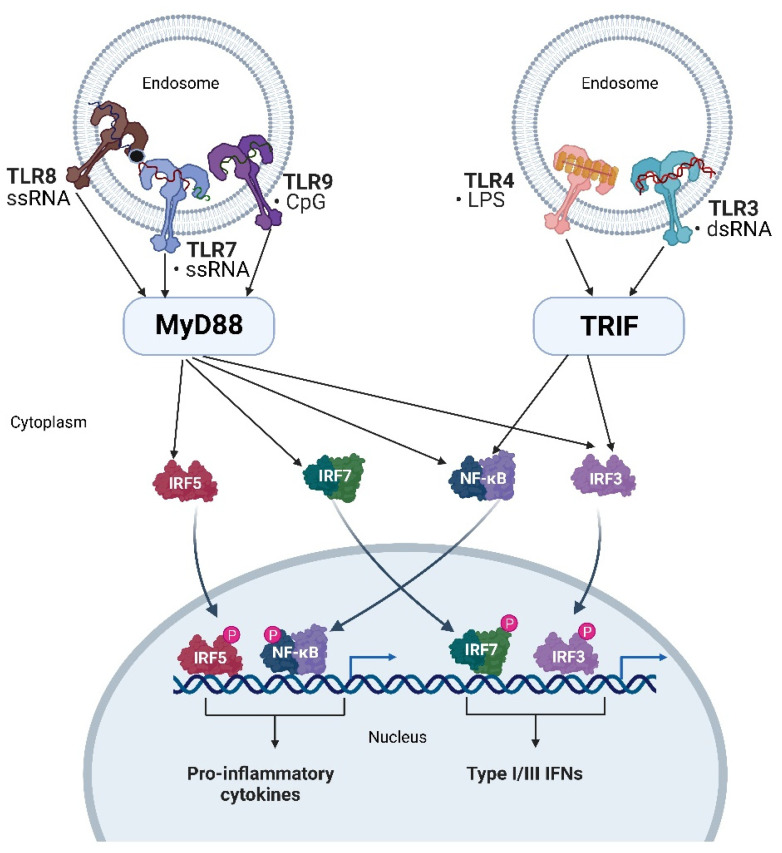
Schematic illustration of TLRs localized to endosomes and the downstream signaling pathways modified from [57]. The expression pattern of endosomal TLRs differs between different cell types and there are also species variations as well as differential up- and downregulation in response to danger signals. TLR4 can localize both to the plasma membrane and the endosomes depending on the cell type in response to ligands such as LPS. TLR3 recognizes dsRNA, TLR7 recognizes ssRNA on one binding site combined with binding to guanosine on the other site, while TLR8 recognizes ssRNA combined with binding to uridine. TLR9 responds to unmethylated CpG motifs that are present at high frequency in bacteria. TLR signaling begins once ligand binding has induced receptor dimerization, which is followed by the engagement of the TIR domain-containing adaptor proteins TRIF (right) or MyD88 (left). The engagement of adaptor proteins will initiate downstream signaling cascades which will lead to phosphorylation and nuclear translocation of transcription factors encoding for proinflammatory cytokines or type I and III interferons. Created with BioRender.com.

**Figure 2 ijms-23-14593-f002:**
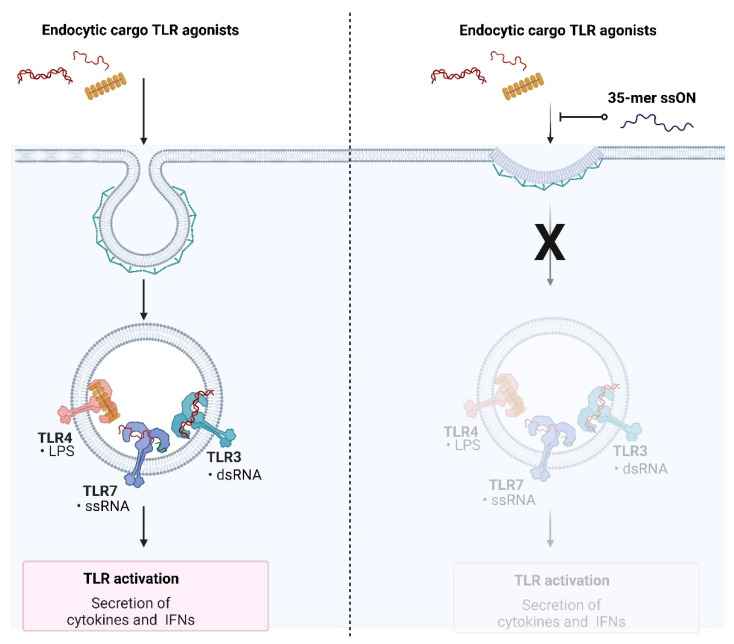
Schematic picture of ssON-mediated interference of endocytosis (SOMIE) modified from [15]. TLR ligands LPS, dsRNA and ssRNA are taken up through clathrin-mediated endocytosis into endosomes expressing TLR4, TLR3, and TLR7 in cells such as human moDC, which will trigger TLR activation and secretion of cytokines and IFNs (**left**). In the presence of 35-mer ssON in the extracellular space, clathrin-mediated endocytosis is temporarily inhibited until ssON is degraded into smaller pieces and consequently endosomal TLR 4/3/7 activation is blocked in moDC (**right**). Created with BioRender.com.

**Figure 3 ijms-23-14593-f003:**
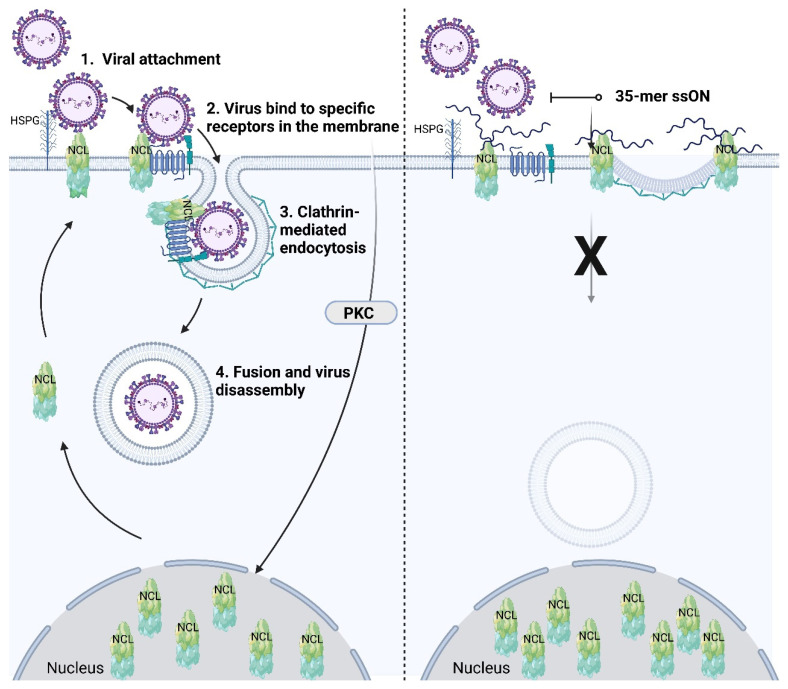
Schematic illustration of nucleolin (NCL) as an attachment receptor for virus and participation in clathrin-mediated endocytosis (**left**), which can be inhibited by 35-mer ssON (**right**). Initially, certain viruses attach to the cellular membrane using heparan sulfate proteoglycan (HSPG) and/or NCL. Next, viruses bind to specific receptors involved in viral entry, and may also trigger intracellular signaling. Hence, viral glycoproteins, such as the RSV F-protein, binds to cellular receptors (IGF1R for RSV) which initiates signaling (PKCζ for IGF1R) [120]. The signaling leads to translocation of NCL from the nucleus to the cell surface. Cell surface NCL also interacts with the C-terminal residues of clathrin light chain A, which is a key component in clathrin-dependent endocytosis. Upon specific receptor binding, the viral entry process is initiated and often occurs through clathrin-mediated endocytosis. Subsequently, viral fusion and virus disassembly may occur. Nucleolin regulates viral attachment and binds nucleic acids as well as aptamer AS1411 through binding of the C-terminal glycine/arginine-rich domain (yellow-green). However, viral proteins (such as RSV F-protein) may also interact with NCL RBD for internalization [113]. The 35-mer ssON but not 15-mer ssON can inhibit viral attachment by shielding NCL [3,4]. We hypothesize that the 35-mer ssONs confer steric hindrance for the endocytosis to occur, while the shorter ssONs are unable to block the molecular movements and are instead readily taken up via clathrin-mediated endocytosis. Created with BioRender.com.

**Table 1 ijms-23-14593-t001:** Characteristics of clinically approved therapeutic oligonucleotides.

Trade Name (Name), Company	Class	Chemistry	Indication, (Target), Organ	FDA/EMA Approval Year	Comment
Vitravene (fomivirsen) Ionis Pharma Novartis	ASO	21-mer PS DNA	CMV renitis, (viral *IE2* mRNA), eye	1998	First approved nucleic acid drug [18]. Withdrawn from use due to reduced clinical need. Mechanism unclear.
Macugen, (pegaptanib) NeXstar Pharma Eyetech/Pfizer	Aptamer	28-mer 2′-F/2′-OMe/pegylated RNA [19,20]	Age-related macular degeneration (VEGF-165), eye	2004	Anti-angiogenic, intravitreal injection.
Kynamro (mipomersen) Ionis Pharma, Genzyme Kastle Tx	Gapmer ASO	20-mer PS 2′-MOE [21]	Homozygous familial hypercholesterolaemia, (*APOB* mRNA), liver	2013	RNase H-mediated cleavage of apolipoprotein B mRNA. Subcutaneous (SC) injection.
Defitelio (defibrotide), Jazz Pharma	Mix of DNA isolated from porcine mucosa	Mix of PO- ssDNA and dsDNA	Hepatic veno-occlusive disease [22], (NA), liver	2016	Sequence-independent mechanism of action. Intravenous injection (IV) [23].
Exondys 51 (eteplirsen), Sarepta Tx	ASO	30-mer PMO	Duchenne muscular dystrophy, (*DMD* exon 51), skeletal muscle	2016	Steric block, splice-switching, [24] IV injection.
Spinraza (nusinersen) Ionis Pharma Biogen	ASO	18-mer PS 2′-MOE	Spinal muscular atrophy, (*SMN2* exon 7), CNS	2016	Steric block, splice-switching [25], Intrathecal injection.
Onpattro (patisiran) Alnylam Pharma	siRNA	19+2^¤^-mer 2′-OMe, ds [26]	Hereditary transthyretin-mediated amyloidosis, (*TTR*), liver	2018	Lipid nanoparticle formulation, IV injection.
Tegsedi (inotersen) Inonis Pharma Akcea Pharma	Gapmer ASO	20-mer PS 2′-MOE [27]	Hereditary transthyretin amyloidosis, (*TTR*), liver	2018	RNaseH mechanism of action, leading to reductions in TTR protein [28], SC injection.
Waylivra (volanesorsen) Ionis Pharma Akcea Pharma	Gapmer ASO	20-mer PS 2′-MOE [29]	Familial chylomicronaemia syndrome, (*APOC3*), liver	2019	Only approved by EMA not FDA. RNaseH mechanism of action, leading to reductions in apoC3 proteins, SC injection.
Givlaari (givosiran) Alnylam Pharma	siRNA	21/23-mer With partial PS, 2′-F, 2′-OMe, ds [30]	Acute hepatic porphyria (*ALAS1*), liver	2019	GalNAc conjugate to target hepatocytes, SC injection.
Vyondys 53 (golodirsen) Sarepta Tx	ASO	25-mer PMO	Duchenne muscular dystrophy, (*DMD* exon 53), skeletal muscle	2019	Splice-switching [31], IV injection.
Viltepso (viltolarsen) NS Pharma	ASO	21-mer PMO	Duchenne muscular dystrophy, (*DMD* exon 53), skeletal muscle	2020	Splice-switching [32], IV injection.
Oxlumo (lumasiran) Alnylam Pharma	siRNA	21/23-mer With partial PS, 2′-F, 2′-OMe, ds [33]	Primary hyperoxaluria type 1 (*HAO1*), liver	2020	GalNAc conjugate to target hepatocytes, SC injection.
Leqvio (inclisiran) Alnylam Pharma Novartis	siRNA	21/23-mer With partial PS, 2′-F, 2′-OMe, ds *	Hypercholesterolemia (*PCSK9*), liver	2021	GalNAc conjugate to target hepatocytes, SC injection.
Amondys 45 (casimersen) Sarepta Tx	ASO	22-mer PMO	Duchenne muscular dystrophy, (*DMD* exon 45), skeletal muscle	2021	Splice-switching, [34] IV injection.
Amvuttra (vutrisiran) Alnylam Pharma	siRNA	21/23-mer With partial PS, 2′-F, 2′-OMe, ds ^	Hereditary transthyretin amyloidosis [35] (*TTR*), liver	2022	GalNAc conjugate to target hepatocytes, SC injection.

Abbreviations: 2′-Ome, 2′-O-methyl; 2′-F, 2′-fluoro; PEG, polyethylene glycol; PO, phosphodiester; PS, phosphorothioate; PMO, phosphorodiamidate morpholino oligomer; 2′-MOE, 2′-O-methoxyethyl; ds, double-stranded; ss, single-stranded; *PCSK9,* proprotein convertase subtilisin kexin type 9; GalNac, N-Acetylgalactosamine; IV, intravenous; SC, subcutaneous. * LEQVIO^®^ (inclisiran) injection, for subcutaneous use (fda.gov). ^ Novel Drug Approvals for 2022|FDA. ^¤^ duplex of two 21-mer RNA with 19 complementary bases and terminal 2-nucleotide 3′overhangs.

**Table 2 ijms-23-14593-t002:** Brief description * of different types of sncRNA.

sncRNA	Length	Comment on Mechanism and/or Function	Ref.
siRNA	20–27 nt	Base pairing and gene silencing with AGO.	[48]
miRNA	21–23 nt	Base pairing and gene silencing with AGO.	[48]
piRNA	21–35 nt	Base pairing and gene silencing with PIWI.	[49]
tsRNA -tRF-1 -tRF-3 -tRF-5 -3′tRNA halves -5′tRNA halves	13–40 nt 13–30 nt 13–30 nt 13–30 nt 30–40 nt 30–40 nt	Gene silencing not always sequence dependent. Tumor suppression, T cell inhibition, affect virus replication and influence stress responses. Transfer RNAs are characterized by their typical cloverleaf structure which can be processed into abundant fragments.	[37]
rsRNA -several names of identified fragments.	Multiple lengths	Shorter 19–24 nt involved in gene silencing with AGO. Longer 74–130 nt function unclear. Often derived from 45S, 5S, and 28S rRNAs in PBMCs with lengths of 15–42.	[37,50]
ysRNA	Multiple lengths often 26–40 nt but also 83–112 nt	Overexpression of a 57 nt increased IL-10 production and administration in vivo in rats conferred cardioprotection. Sequencing in human PBMCs revealed abundant ysRNAs of approx. 26–40 nt. Often derived from YRNA-RNY4 and YRNA-RNY1 in PBMCs	[50,51]
snsRNA	Approx. 16–40 nt	SnRNA is as a family of highly conserved ncRNAs located in the nucleus and associated with Sm ribonucleoproteins and other specific proteins, to form small nuclear ribonucleoproteins. The function of fragments is largely unknown.	[39,52]
snosRNA	Often 20–24 nt but also 17–19 nt and 27–33 nt	Some were shown to be similar to miRNA and can use AGO for gene silencing.	[52,53]
vtsRNA	Full-length vRNA is approx. 100 nt and can be processed into approx. 23 nt	RNA components of Vault ribonucleoprotein particles, which are located in the cytoplasm. Control of apoptosis and autophagy, lysosome biogenesis, and function in cancer cells.	[54]

***** The list and definition of different sncRNAs will require substantial revisions by the field of experts in the sncRNA field and this table is a very rough overview to put the universe of small (s)-derived non-coding RNA into perspective. With improved sequencing including multiple lengths, the picture may change. Peripheral blood mononuclear cells: PBMC.

## Data Availability

Not applicable.

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
