# Peer review of "Is There a Role for Immunoregulatory and Antiviral Oligonucleotides Acting in the Extracellular Space? A Review and Hypothesis"

_ijms, 2022, doi:10.3390/ijms232314593_

Round 1
Reviewer 1 Report
This interesting review discuss the role of small non coding RNA in mediating antiviral effects and modulating innate immune response. The review is well written and highly informative on new type of ssON.
Line 102 SEREX or SELEX
The section on TLRs (section 4) is in more detail and needs to be reduced since this aspect has been reviewed extensively in multiple articles. The authors can remove or trim general details related to TLR, the activation mechanisms and concentrate on how ssON can be utilized to regulate these innate receptors
The section 7 may be renamed as ssONs interfering with nucleolin or similar to that of section 6
Diagrammatic representations for section 6, 7, and 8 may help in easier understanding
Author Response
Reviewer 1. “This interesting review discuss the role of small non-coding RNA in mediating antiviral effects and modulating innate immune response. The review is well written and highly informative on new type of ssON.” Reply: Thank you for the kind feedback.
Line 102 SEREX or SELEX. Reply: misspelling corrected
“The section on TLRs (section 4) is in more detail and needs to be reduced since this aspect has been reviewed extensively in multiple articles. The authors can remove or trim general details related to TLR, the activation mechanisms and concentrate on how ssON can be utilized to regulate these innate receptors.”
Reply: We completely agree with this comment and section 4 was now trimmed.
“The section 7 may be renamed as ssONs interfering with nucleolin or similar to that of section 6”
Reply: thank you for the suggestion-a new title is included.
“Diagrammatic representations for sections 6, 7, and 8 may help in easier understanding.”
Reply: Again, a very nice suggestion, and a new graphical figure is included.
Reviewer 2 Report
This appears to be a very well-organized and well-written review that nicely summarizes several areas of oligo therapeutic knowledge and progress. While the review does not definitively answer its own question from the title, it addresses what is known about oligos in the extracellular space and stimulates reader thought in relevant directions. I only have one minor correction. Line 102 - should read "SELEX" not "SEREX."
Author Response
Reviewer 2.“This appears to be a very well-organized and well-written review that nicely summarizes several areas of oligo therapeutic knowledge and progress. While the review does not definitively answer its own question from the title, it addresses what is known about oligos in the extracellular space and stimulates reader thought in relevant directions. I only have one minor correction. Line 102 - should read "SELEX" not "SEREX.”
Reply: Thank you for this kind feedback-the goal was indeed to stimulate the thoughts of the reader. The misspelling of SELEX is corrected.
